# Treatment Outcomes of Childhood TB Patients in Four TB High Burden States of Malaysia: Results from a Multicenter Retrospective Cohort Study

**DOI:** 10.3390/antibiotics11111639

**Published:** 2022-11-16

**Authors:** Rabbiya Ahmad, Syed Azhar Syed Sulaiman, Abdul Razak Muttalif, Nafees Ahmad, Aseel Rezeq Ali Yaghi, Khang Wen Goh, Long Chiau Ming, Nehad Jaser Ahmed, Amer Hayat Khan

**Affiliations:** 1Department of Clinical Pharmacy, School of Pharmaceutical Sciences, Universiti Sains Malaysia, Penang 11800, Malaysia; 2Department of Medicine, Faculty of Medicine, Bioscience and Nursing, MAHSA University, Jenjarom 42610, Malaysia; 3Department of Pharmacy Practice, Faculty of Pharmacy and Health Sciences, University of Balochistan, Quetta 87330, Pakistan; 4Faculty of Data Science and Information Technology, INTI International University, Persiaran Perdana BBN, Putra Nilai, Nilai 71800, Malaysia; 5Pengiran Anak Puteri Rashidah Sa’adatul Bolkiah, Institute of Health Sciences, Universiti Brunei Drussalam, Gadong BE1410, Brunei; 6Clinical Pharmacy Department, College of Pharmacy, Prince Sattam Bin Abdulaziz University, Al-Kharj 16278, Saudi Arabia

**Keywords:** childhood TB, treatment outcomes, Malaysia, multicenter study

## Abstract

Data regarding treatment outcomes among childhood TB patients are lacking in Malaysia. The present study aimed to evaluate the treatment outcomes and predictors of unsuccessful treatment outcomes among childhood TB patients in four TB high-burden states of Malaysia. This was a retrospective cohort study conducted at 13 healthcare centers in four states of Malaysia, namely, Sabah, Sarawak, Selangor, and Penang. During the study period, a total of 8932 TB patients were enrolled for treatment at the study sites, of whom 206 (2.31%) were children. The majority of the childhood TB patients were female (52.9%) and belonged to the age group of 6–10 years (42.7%). Pulmonary TB accounted for 70.9% of childhood TB. Among childhood PTB patients, 50% were sputum smear negative. One hundred and seventy-eight patients (86.4%) were successfully treated (87 were cured and 91 completed treatment). Among 28 (13.6%) patients with unsuccessful treatment outcomes, 13 (6.3%) died, 3 (1.5%) failed treatment, 9 (4.4%) defaulted, and 3 (1.5%) were transferred out. Multivariate analysis revealed that patients’ age (5–14 years) (OR = 0.279, *p*-value *=* 0.006) and male gender (OR = 0.390, *p*-value *=* 0.046) had a statistically significant negative association with unsuccessful treatment outcomes. The prevalence of childhood TB in the current study was comparable to the recently published national estimates. The study sites reached the WHO target of treatment success. Special attention to patients with identified risk factors can improve treatment outcomes.

## 1. Introduction

In 2019, an estimated 10 million tuberculosis (TB) cases occurred globally [1]. Globally, children with TB accounted for 11% of the total TB cases in 2018. In Malaysia, children had approximately 3% of the total TB cases for the 2010–2015 Malaysian cohort. In terms of TB burden, Malaysia was considered to have an intermediate TB burden with an estimated incidence rate of 92 cases per 100,000 people in 2018 [2]. A study of the global estimation of TB burden among children revealed that 239,000 patients died in 2015, of whom 80% were under 5 years old, and more than 96% did not receive any prior TB treatment [3]. Childhood TB (TB in patients aged <15 years) is an indicator of recent transmission in the community and is suggestive of a poor TB control program [4]. WHO recommends that childhood TB be reported and notified through national tuberculosis control programs (NTPs). Due to difficulties in diagnosis, nonspecific symptoms, poor recording and reporting practices, and lack of research and investment, childhood TB is suffering from historical neglect, and treatment outcomes in childhood TB are rarely evaluated by NTPs [5].

Malaysia is a Southeast Asian country and is included in the Western Pacific Region of WHO. In 2019, the estimate for the TB incidence rate in Malaysia was 92 cases per 100,000 population, and the TB mortality rate was estimated at 4 cases per 100,000 population per year [6]. Just like in other parts of the world, data regarding TB treatment outcomes and predictors of unsuccessful outcomes in childhood TB are lacking in Malaysia. Thus, the present study was conducted to evaluate the treatment outcomes and predictors of unsuccessful treatment outcomes among childhood TB patients in four TB high-burden states in Malaysia.

## 2. Results

Over 3 years, a total of 8932 TB patients were enrolled for treatment at the study sites, of whom 206 (2.31%) were children. The median age and weight of childhood TB patients at the baseline visit were 8.36 + 4.20 years and 30.63 + 10.82 kg, respectively. The majority of the childhood TB patients were females (52.9%), belonged to the age group of 6–10 years (42.7%), were urban residents (51.5%), and were registered as new TB patients (94.2%). Twenty-six (12.6%) patients suffered from concurrent comorbidity. Pulmonary TB accounted for the majority (70.9%) of childhood TB cases. Among childhood patients with PTB, 50% were sputum smear negative, and 45.5% of the patients had lung lesions on baseline chest X-rays. No statistically significant association was observed between any variable and sputum smear-positive pulmonary TB. The baseline sociodemographic and clinical characteristics of the study participants are given in Table 1.

### 2.1. Tuberculosis Treatment Outcomes

One hundred and seventy-eight patients (86.4%) were successfully treated. Among them, 87 were cured and 91 completed treatment. Among the 28 (13.6%) patients with unsuccessful treatment outcomes, 13 (6.3%) died, 3 (1.5%) failed treatment, 9 (4.4%) defaulted, and 3 (1.5%) were transferred out (Table 2).

### 2.2. Predictors of Unsuccessful Treatment Outcomes

In univariate analysis, male gender (OR = 0.400, *p*-value = 0.039), age of 5–14 years (OR = 0.315, *p*-value = 0.007), and presence of comorbidity (OR = 2.789, *p*-value = 0.040) had statistically significant association with unsuccessful treatment outcomes (Table 3). 

Multivariate analysis revealed that patients’ age (5–14 years) (OR = 0.279, *p*-value = 0.006) and male gender (OR = 0.390, *p*-value = 0.046) had statistically significant negative association with unsuccessful treatment outcomes. Patients with male gender and belonging to the age group of 5–14 years were significantly less likely than their counterparts to develop unsuccessful treatment outcomes. This model fit was based on a nonsignificant Hosmer–Lemeshow test (*p*-value = 0.620) and an overall percentage of 86.4% from the classification table (Table 4).

## 3. Discussion

Although various studies from Malaysia have evaluated the treatment outcomes and predictors of unsuccessful outcomes among TB patients, none has focused on childhood TB patients as a separate cohort [7,8]. To the best of our knowledge, the present study is the first from the country to report treatment outcomes and predictors of unsuccessful treatment outcomes among childhood TB patients. In the current study, the proportion of childhood TB among the registered TB patients at the study sites was 2.31%. This was comparable to the recently reported national figure of childhood TB in Malaysia (2.87%) but was lower than the global figure (6.5%) and the proportion of childhood TB patients in the Western Pacific Region of WHO (3.4%) [9]. The fractionally greater proportion of childhood TB patients in the country reported by the latest global TB report could be attributed to the improved TB case detection rate (57% in 2008 vs. 78% in 2014) [9] and reporting and recording system. Childhood TB is a marker of recent transmission in the community, and its high prevalence indicates the failure of public health prevention measures for NTP [4]. There are certain modern diagnostic methods that allow for rapidly diagnosing the disease and, at the same time, evaluating the molecular epidemiology and dynamics of TB transmission, including resistant forms. Whole-genome sequencing (WGS) is a viable and financially feasible tool for a timely and comprehensive diagnosis of drug resistance in developed countries [10]. The comparatively lower percentage of childhood TB patients in the current study (2.31%) and the national figure reported by the recently published global TB report (2.87%) suggest the effectiveness of public health prevention measures for Malaysian NTP. 

The risk of developing active TB after being infected with MTB is determined by various factors, including age, virulence of the organism, immune status, and magnitude of initial infection [11]. Generally, younger children, particularly those <5 years of age, are at greater risk of developing active TB [11,12]. However, in the current cohort of patients, children aged <5 years made up only 23.8% of the registered childhood TB patients. Similar results regarding a comparatively lower proportion of childhood PTB patients aged <5 years have been reported in Addis Ababa and Thailand [13]. The possible reason for the lower proportion of childhood TB patients aged <5 years in the current study could be the underdiagnosis of TB in this group of patients [13]. As children in the age group of <5 years generally have comparatively less interaction with the community than children of school-going age (>5 years), this could be the other possible reason for a lower proportion of TB prevalence in children <5 years of age.

A notable proportion of study participants (22.3%) suffered from EPTB. This was comparable to the proportion of childhood EPTB patients reported by WHO (>20%) and studies conducted in Southern Ethiopia (24.8%) [7] and the USA (17%) [14], but lower than that observed in studies conducted in Addis Ababa (40%) [13], Thailand (29%), and Southern Taiwan (27.5%) [15]. However, the relatively higher proportion of EPTB patients in the current study than the recently reported national figure of EPTB patients in Malaysia (13.1%) warrants the need for an epidemiological study on childhood TB in the country. In the current cohort of childhood patients, TB lymphadenitis accounted for 78.3% of EPTB. This finding was in agreement with the accepted epidemiology of TB lymphadenitis as the highly prevalent form of EPTB in children and adults around the globe. Half of the children with PTB were sputum negative. As childhood TB patients usually suffer from primary TB rather than secondary, have a paucibacillary load, and are unable to expectorate sputum [5], this finding seems logical. Likewise, a high proportion of sputum smear-negative childhood PTB patients have been observed in studies conducted in Malawi, Thailand, and Addis Ababa [13].

More than 86% of the study participants achieved successful treatment outcomes, and the study sites collectively reached the WHO target of treatment success rate (>85%). This was well above the recently reported national figure of TB treatment success rate (76%) and the success rate reported for the 2012 cohort of TB patients of all ages (78.5%) registered in the Malaysian national TB registry [7]. Likewise, the treatment success rate among childhood TB patients (85.5%) has been reported by a study conducted in Addis Ababa. However, the treatment success rate in the current study was comparatively higher than that observed among childhood TB patients in studies conducted in Malawi (54%) [16], Botswana (67%) [17], Thailand (72%) [13], and Lagos, Nigeria (79.2%) [18]. As the treatment default is often the main contributing factor for poor treatment outcomes in adult TB patients [19,20], the comparatively high treatment success rate in the current study could be attributed to a comparatively low default rate (4.4%) than that observed in studies conducted in Thailand, Malawi, and Lagos, Nigeria (>12%) [7]. Possible differences in parents’ level of education and awareness about TB and programmatic efforts at the current study sites to trace TB patients with delays in scheduled visits on the phone and [21] home visits by nurses to ensure patients’ adherence to the treatment could be possible reasons for the relatively low default rate. In the current cohort, the mortality rate observed (6.3%) was lower than that recently reported for TB patients of all ages from Malaysia (9.3%), but comparable to studies conducted among childhood TB patients in Lagos, Nigeria (6%) [16], Thailand (6%) [15], and Southern Ethiopia (5.3%) [22]. However, studies in Malawi (17%) and Botswana (10.5%) [23] have reported comparatively higher mortality rates among childhood TB patients [23].

In multivariate analysis, female sex and patients aged <5 years emerged as risk factors for unsuccessful treatment outcomes. Unlike our finding, no significant association between the gender of childhood TB patients and treatment outcomes has been reported by studies conducted elsewhere. Because of the limited number of treatment default and transferred-out patients in the current study, we were unable to evaluate the risk factors associated with these two outcome categories. As the mortality (females = 7.3% vs. males = 5.2%) and treatment failure rates (females = 1.8% vs. males = 1%) of male and female patients in the current study were somewhat similar, the high default (females = 1.8% vs. males = 1%) and transferred-out rates (females = 2.7% vs. males = 0%) observed among female patients (Table 2) contributed largely to make female gender a significant risk factor for unsuccessful treatment outcomes. A study with a large sample size is needed to confirm the present finding and evaluate the risk factors associated with TB treatment default in childhood patients. In the current study, patients aged <5 years emerged as a risk factor for unsuccessful treatment outcomes. The significant association between patients aged <5 years and unsuccessful treatment outcomes was largely due to the higher mortality (19%) and default rates (4.3%) among patients <5 years of age as compared with the death (2.5%) and default rates (0.6%) among patients 5–14 years of age (Table 2). Similar findings regarding childhood TB patients aged <5 years and unsuccessful treatment outcomes (death and default) have been reported by studies [7] conducted elsewhere [18]. The immature immune system in childhood TB patients <5 years of age is the possible explanation for the high mortality rate in this group of patients [6].

Although it was a multicenter study conducted in four TB high-burden states of Malaysia, the convenient selection of the study sites might limit the generalizability of the current findings. The sample size included a small number of children aged less than 5, which can lead to study bias; hence, larger population study should be performed to generalize the results. In addition, we were unable to evaluate certain socioeconomic data, such as monthly family income, parents’ educational level and awareness about TB, and occurrence of ADRs in the study participants, which might have an impact on the observed treatment outcomes. 

## 4. Materials and Methods

### 4.1. Study Design and Settings

This was a retrospective cohort study conducted in 13 healthcare centers in four TB high-burden states of Malaysia, namely, Sabah (Queen Elizabeth Hospital, Luyan Health Clinic, Sandakan Hospital, and Tawau Hospital), Sarawak (Anti-TB Association Clinic, Kuching Hospital, Sibu Hospital, Miri Hospital, Limbang Hospital, and Sri Aman Hospital), Selangor (Kajang Hospital and Sungai Buloh Hospital), and Penang (Penang General Hospital). 

### 4.2. Diagnosis and Treatment of TB in Children at the Study Sites

At the study sites, the presence of at least three of the following key features was considered strong indicators of TB presence in the study participants: (i) chronic symptoms suggestive of TB, (ii) physical signs highly suggestive of TB, (iii) a positive tuberculin skin test (TST), and (iv) chest X-ray suggestive of TB. Children suspected of pulmonary TB underwent sputum smear examination and chest X-ray. Those patients who were unable to expectorate sputum underwent gastric lavage/aspiration. Patients with (i) two or more sputum smear-positive examinations for acid-fast bacilli (AFB) or (ii) one sputum smear-positive result for AFB plus chest X-ray findings suggestive of active PTB or (iii) one sputum smear and one sputum culture-positive result for AFB were enrolled as smear-positive pulmonary TB (PTB) patients. PTB patients without positive sputum smear results for AFB were enrolled as sputum smear-negative PTB patients. Clinical presentation, specimen smear, culture, and histopathology were used to diagnose extrapulmonary TB (EPTB) [24]. 

Patients registered under category I (new smear-positive PTB, new smear-negative PTB with extensive lung involvement, severe EPTB, and patients with HIV comorbidity) were treated for 2 months (intensive phase, IP) with isoniazid (H), rifampicin (R), pyrazinamide (Z), and ethambutol (E) and for 4 months (continuation phase, CP) with HR. Patients registered under category III (new smear-negative PTB other than category I, less severe forms of EPTB) were treated with HRZ in IP and HR in CP. Patients registered under category II (patients with a history of previous TB treatment) were treated for 2 months with HRZE + streptomycin (S), 1 month with HRZE, and 5 months with HRE [24]. 

### 4.3. Study Population and Data Collection

All eligible TB patients (aged < 15 years) enrolled for treatment at the study sites from 1 January 2006 to 31 December 2008 were included in the study. Patients were followed until treatment outcome was reported. A purpose-developed validated data collection form was used to extract the patients’ sociodemographic, clinical, and microbiological data from their medical records. Sociodemographic data included patients’ age, gender, race, baseline body weight, and residence. Clinical data included patients’ clinical presentation at the baseline visit, site of TB, registration category, chest X-ray findings, and comorbid conditions. Microbiological data included sputum smear results of smear-positive pulmonary TB patients at baseline visits and subsequently at 2 and 6 months of treatment. In addition, sputum and other specimen smear and culture results at the baseline visit were also collected. Diagnosis of comorbidities was based on patients’ medical records. Treatment outcomes were defined as per WHO guidelines criteria [24]. Cured and treatment-completed outcomes were grouped as successful treatment outcomes, whereas death, default, and transferred out were grouped as unsuccessful treatment outcomes.

### 4.4. Treatment Outcomes

WHO-recommended regimens are largely used for program evaluation and decision making in the context of TB programmatic management. WHO-recommended criteria are distinct from research-based definitions, which are primarily used to assess pharmacological and therapy regimens. All new and retreatment PTB patients report their treatment outcomes according to WHO and NTP guidelines. The definition of treatment outcomes is adapted from the Definitions and Reporting Framework for Tuberculosis, 2013 revision, World Health Organization, 2013, Geneva, Switzerland [25]:

Cured: A patient with bacteriologically confirmed TB at the beginning of treatment who was smear or culture negative in the last month of treatment and on at least one previous occasion.

Treatment completed: A TB patient who completed treatment without evidence of failure but with no record to show that sputum smear or culture results in the last month of treatment and on at least one previous occasion were negative, either because tests were not done or because results are unavailable.

Died: A TB patient who dies for any reason before starting or during treatment.

Loss to follow-up: A TB patient who did not start treatment or whose treatment was interrupted for 2 consecutive months or more.

Treatment failed: Tb patient whose sputum smear or culture was positive at 5 months or later during TB treatment.

Not evaluated: A TB patient for whom no treatment outcome is assigned. This includes cases of “transferred out” to another treatment unit and cases for whom the treatment is unknown to the reporting unit.

### 4.5. Statistical Analysis

Statistical Package for Social Sciences (SPSS), version 16, was used for data analysis. Continuous data were presented as means + SD, medians, and ranges. Categorical data were presented as frequencies and percentages. Multivariate logistic regression analysis with Wald statistical criteria was used to obtain the final model for predictors of unsuccessful treatment outcomes. Independent variables with *p*-value < 0.2 in univariate analysis were entered in multivariate analysis [9]. Statistical significance was taken at *p*-value < 0.05. 

This study was approved by the Clinical Research Centre (CRC) of the Penang General Hospital.

## 5. Conclusions

Despite the limitations associated with the current study, several conclusions can be drawn from it. The proportion of childhood TB patients in the study areas was comparable to the reported national figure of childhood TB patients in the country. The study sites met the WHO target of a TB treatment success rate of >85%. To further improve the TB treatment success rate among childhood TB cases, special attention should be paid to patients of female gender and aged <5 years.

## Figures and Tables

**Table 1 antibiotics-11-01639-t001:** Patients’ baseline sociodemographic and clinical characteristics.

Variable	Mean ± SD	No. (%)
**Gender**		
Female		109 (52.9)
Male		97 (47.1)
**Age (years)**	8.36 ± 4.20	
0–5		49 (23.8)
6–10		88 (42.7)
11–14		69 (33.5)
**Weight (kg)**	30.63 ± 10.82	
**State**		
Penang		14 (6.8)
Sabah		142 (68.9)
Sarawak		39 (18.9)
Selangor		11 (5.3)
**Race**		
Malay		40 (19.4)
Chinese		38 (18.4)
Indian		7 (3.4)
Sabahan		70 (34.0)
Sarawakian		19 (9.2)
Others		32 (15.5)
**Residence**		
Urban		106 (51.5)
Rural		100 (48.5)
**Previous TB treatment**		
No		194 (94.2)
Yes		12 (5.8)
**Site of TB**		
PTB		146 (70.9)
PTB + EPTB		14 (6.8)
Sputum smear-positive PTB		80 (50.0)
Sputum smear-negative PTB		80 (50.0)
EPTB		46 (22.3)
Tuberculous lymphadenitis		36 (17.5)
Pleural TB		9 (4.4)
CNS TB		12 (5.8)
Abdominal TB		1 (0.5)
Genitourinary TB		1 (0.5)
Miliary TB		1 (0.5)
**Comorbidity**		
No		180 (87.4)
Yes		26 (12.6)
**Type of comorbidity**		
HIV		8 (3.9)
DM		2 (1.0)
Hepatitis		15 (7.3)
Others		5 (2.4)

DM, diabetes mellitus; HIV, human immunodeficiency virus; kg, kilogram; SD, standard deviation TB, tuberculosis.

**Table 2 antibiotics-11-01639-t002:** Treatment outcomes of study participants.

Variable	Treatment Outcomes No. (%)					
	Successful		Unsuccessful			
	Cured	Completed	Died	Failed	Defaulted	Transfer Out
**Gender**						
Female	45 (41.3)	44 (40.0)	8 (7.3)	2 (1.8)	7 (6.4)	3 (2.7)
Male	42 (43.3)	47 (48.5)	5 (5.2)	1 (1.0)	2 (2.1)	-
**Age (years)**						
0–4	2 (4.3)	32 (69.6)	9 (19.6)	-	2 (4.3)	1 (2.2)
4–15	85 (53.1)	59 (36.9)	4 (2.5)	3 (1.9)	1 (0.6)	2 (1.2)
**Race**						
Malay	16 (40.0)	16 (40.0)	4 (10.0)	1 (2.5)	2 (5.0)	1 (5.0)
Chinese	15 (39.5)	20 (52.6)	2 (5.3)	-	1 (2.6)	-
Indian	3 (42.9)	3 (42.9)	-	-	1 (14.1)	-
Sabahan	31 (44.3)	31 (44.3)	3 (4.3)	1 (1.4)	2 (2.8)	2 (2.8)
Sarawakian	9 (38.9)	8 (50.0)	-	1 (5.6)	1 (5.6)	-
Others	15 (46.9)	11 (34.4)	4 (12.5)	-	2 (6.2)	-
**Residence**						
Urban	41 (38.7)	51 (48.1)	9 (8.5)	-	4 (3.8)	1 (0.9)
Rural	46 (46.0)	40 (40.0)	4 (4.0)	3 (3.0)	5 (5.0)	2 (2.0)
**Site of TB**						
Pulmonary TB	82 (56.2)	44 (30.1)	6 (4.1)	3 (2.1)	8 (5.5)	3 (2.0)
EPTB	-	39 (84.8)	7 (15.2)	-	-	-
PTB + EPTB	5 (35.7)	8 (57.1)	-	-	1 (7.1)	-
**Comorbidity**						
No	75 (41.7)	84 (46.7)	11 (6.1)	3 (1.7)	5 (2.8)	2 (1.1)
Yes	12 (46.2)	7 (26.9)	2 (7.7)	-	4 (15.4)	1 (3.8)
**Number of TB symptoms**						
≤2	48 (46.2)	41 (39.4)	5 (4.8)	2 (1.9)	6 (5.8)	2 (1.9)
3–4	32 (38.1)	42 (50.0)	7 (8.3)	1 (1.2)	2 (2.4)	-
≥5	7 (38.9)	8 (44.4)	1 (5.6)	-	1(5.5)	1 (5.5)

PTB, pulmonary TB; EPTB, extrapulmonary TB; TB, tuberculosis.

**Table 3 antibiotics-11-01639-t003:** Univariate analysis of predictors of unsuccessful treatment outcomes.

Variable	Treatment Outcome No. (%)		OR (95% CI)	*p*-Value
	Unsuccessful	Successful		
**Gender**				
Female	20 (18.3)	89 (81.7)	Referent	
Male	8 (8.2)	89 (91.8)	0.400 (0.167–0.956)	**0.039**
**Age (years)**				
0–4	12 (26.1)	34 (73.9)	Referent	
4–15	16 (10.0)	144 (90.0)	0.315 (0.136–0.727)	**0.007**
**Race**				
Malay	8 (20.0)	32 (80.0)	Referent	
Chinese	3 (7.9)	35 (92.1)	0.343 (0.084–1.406)	**0.137**
Indian	1 (14.3)	6 (85.7)	0.667 (0.070–6.353)	0.724
Sabahan	6 (18.8)	26 (81.2)	0.923 (0.284–2.999)	0.894
Sarawakian	2 (10.5)	17 (89.5)	0.500 (0.095–2.634)	0.414
Others	8 (11.4)	62 (88.6)	0.516 (0.177–1.503)	0.225
**Residence**				
Urban	14 (13.2)	92 (86.8)	Referent	
Rural	14 (14.0)	86 (86.0)	1.070 (0.482–2.734)	0.868
**Site of TB**				
Pulmonary TB	20 (13.7)	126 (86.3)	Referent	
EPTB	7 (15.2)	39 (84.8)	1.131 (0.445–2.874)	0.796
PTB + EPTB	1 (7.1)	13 (92.9)	0.497 (0.060–3.910)	0.497
**Comorbidity**				
No	21 (11.7)	159 (88.3)	Referent	
Yes	7 (26.9)	19 (73.1)	2.789 (1.048–7.424)	**0.04**
**Number of TB symptoms**				
≤2	15 (14.4)	89 (85.6)	Referent	
3–4	10 (11.9)	74 (88.1)	0.802 (0.304–1.890)	0.614
≥5	3 (16.7)	15 (83.3)	1.187 (0.306–4.600)	0.804

PTB, pulmonary TB; EPTB, extrapulmonary TB; TB, tuberculosis.

**Table 4 antibiotics-11-01639-t004:** Multivariate analysis of predictors of unsuccessful treatment outcomes.

Variable	B	SE	OR (95% CI)	*p*-Value
Male gender	−0.941	0.472	0.390 (0.155–0.985)	0.046
Age 5–14 years	−1.276	0.465	0.279 (0.112–0.965)	0.006
Chinese race	−8.93	0.752	0.410 (0.094–1.789)	0.235
Comorbidity	0.930	0.560	2.535 (0.863–7.452)	0.090

B, beta; OR, odds ratio; SE, standard error.

## Data Availability

Data will be provided upon demand.

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
