# Peer review of "Treatment Outcomes of Childhood TB Patients in Four TB High Burden States of Malaysia: Results from a Multicenter Retrospective Cohort Study"

_antibiotics, 2022, doi:10.3390/antibiotics11111639_

Round 1

Reviewer 1 Report

1. This sentence mentioned more than once in the paper is very confusing “One hundred and seventy-eight patients (86.4%) were successfully treated. Among them, 87 were cured and 91 completed treatment”. Does this mean that out of 178 only 91 completed full treatment and 87 didn’t, but all 178 were cured of TB? To me both “successful treatment” and “cured” means the same thing. Authors should clarify and state it accordingly. Accordingly modify Table 2.

2. Authors mention that in the current cohort of patients, children with age <5 years made up only 23.8% of the registered children TB patients. Is it possible that the variate analyses were biased? In other words, is it possible that with a larger sample size authors could have reached a different conclusion? Is yes it needs to be stated in the discussion.

3. The subheadings of Table 2 and 3 are on top of each other for eg. Cured & Completed and Unsuccessful & Successful. Please space them out.

4. Line 25 rephrase “the to” to “to the”.

5. Line 112 rephrase “of risk” to “risk of”.

Reviewer 2 Report

This retrospective cohort study collected the data of 206 childhood TB patients which evaluated the predictors of successful treatment outcomes in four study sites of Malaysia.This manuscript presented some important information for the treatment outcomes of the pediatric TB patients in Malaysia. However, there are some problems to be further improved as well:

Major:

1, Factors like BMI and nutritional status, that could affect treatment outcomes were not recorded and thus not examined.

2, There is not any data about close contacts of these patients. How can the authors make a conclusion about the close contacts?

3.Data on drug resistance test could be added.

4. How is the adherence of the children? It is significantly associated with outcome.

Minor:

1, There are some spelling errors in the manuscript, such as, in table 1, plural would be pleural, on page 5, in line 112, The of riskwould be the risk of. Please check the manuscript carefully.

2, It would be better to cite the latest global tuberculosis report in the new manuscript.

Reviewer 3 Report

Tuberculosis in children is very important and needs to be monitored, especially in high-incidence TB countries.  In this retrospective study enrolling 8932 TB patients, only 2.33% were children. Based on the suggestions below, I highly encourage authors to make a minor revisions before publishing:

#line 34 – remove the green colour

#line 37 – please update the information about the incidence (including the children and the situation in Malaysia) according to the latest WHO report

#line 45 – please do not use but on the beginning of the sentence

Did you performed the phenotypic of genotypic drug susceptibility testing for isolates studied? This may be also a crucial factor influencing the treatment outcome in patients with TB.

The results showed that almost half of the children with PTB were sputum negative. Have you considered blood diagnostics (eg QuantiFERON-TB)? Its fast and without the need for sputum sample.

You mentioned following statement in the discussion section : Childhood TB is a marker of recent transmission in the community and its high prevalence indicates the failure of public health prevention measures for NTP. However, I am missing possible solutions. I highly encourage the authors to include in this section some modern diagnostic methods, including WGS, which allow to rapidly diagnose the disease and at the same time to evaluate the molecular epidemiology and dynamics of TB transmission, including resistant forms (references: doi.org/10.3389/fmicb.2019.01741; doi.org/10.1038/s41598-022-11287-5; doi.org/10.1128/JCM.01582-19).

Table 1. please edit the legend

#line 34 – please follow the uniform expression of numbers

You have any information about the incidence of children TB patients nowdays?

Did you observed any differences in treatment outcomes comparing patients in Category I vs Category II vs Category III?
